# Iron-Coupled Anaerobic Oxidation of Methane in Marine Sediments: A Review

**Hailin Yang** , **Shan Yu** * **and Hailong Lu** *

Beijing International Center for Gas Hydrate, School of Earth and Space Sciences, Peking University, Beijing 100871, China; hyang@pku.edu.cn
* Correspondence: hlu@pku.edu.cn (H.L.); shanyu@pku.edu.cn (S.Y.)

**Abstract:** Anaerobic oxidation of methane (AOM) is one of the major processes of oxidizing methane in marine sediments. Up to now, extensive studies about AOM coupled to sulfate reduction have been conducted because $SO_4^{2-}$ is the most abundant electron acceptor in seawater and shallow marine sediments. However, other terminal electron acceptors of AOM, such as $NO_3^-$, $NO_2^-$, Mn(IV), Fe(III), are more energetically favorable than $SO_4^{2-}$. Iron oxides, part of the major components in deep marine sediments, might play a significant role as an electron acceptor in the AOM process, mainly below the sulfate–methane interface, mediated by physiologically related microorganisms. Iron-coupled AOM is possibly the dominant non-sulfate-dependent AOM process to consume methane in marine ecosystems. In this review, the conditions for iron-coupled AOM are summarized, and the forms of iron oxides as electron acceptors and the microbial mechanisms are discussed.

**Keywords:** anaerobic oxidation of methane; iron reduction; microbial mechanism; marine sediments; carbon sink

## 1. Introduction

Methane is the second most powerful anthropogenic greenhouse gas in the atmosphere, after carbon dioxide, and it plays an important role in marine and atmospheric chemistry [1]. Considered on an equivalent mass basis, methane, which has a relative global warming potential of 265, 34 times higher than that of $CO_2$ present in the atmospheric environment, contributes to 16% of global warming [2–4]. Many chemical reactions in the atmosphere, especially in the stratosphere and troposphere, are controlled by the atmospheric $CH_4$ [5]. Oceanic methane has an impact on the chemistry and biology of both sediments and the overlying water column, and the oxidation of methane is a major process by which organic matter is recycled back into the ocean [6]. In anoxic marine sediments, about 85~300 Tg methane is produced annually, accounting for up to 30% of the global methane output [1,7]. In comparison, marine environments only contribute 2% of the total global methane emission because up to 90% of the methane is consumed by the anaerobic oxidation of methane (AOM) process conducted by microorganisms and not escaping into the atmosphere [1].

AOM is a microbial process and mainly occurs in a vertical geochemical region referred to as the sulfate–methane transition zone (SMTZ), where the upward migrating methane encounters the downward diffusing sulfate. Based on the electron acceptors involved, AOM can be categorized into several types. Due to a large amount of sulfate in the oceanic ecosystem, sulfate reduction coupled AOM (as shown in Equation (1)), which can be conducted only via microbes, is considered the major AOM process in marine sediments and has been a subject of intense investigation for decades [8,9]. However, other available terminal electron acceptors coupled to AOM, such as $NO_3^-$, $NO_2^-$, and oxides of iron or manganese could provide more free energy by oxidizing methane anaerobically rather than sulfate (Figure 1). Iron is the fourth most abundant element in the Earth's crust. Every year, a massive amount of iron is supplied to oceans from rivers, and consequently, iron-containing minerals are part of the major components in natural sediments. Iron

oxides are thermodynamically favorable electron acceptors for AOM. For instance, ferrihydrite coupled AOM yields almost thirty times as much free energy as AOM with sulfate (Equation (2)) [10]. All the above implies that iron-coupled AOM (Fe-AOM) might play an important role in marine methane oxidization and might have been underestimated, especially in sulfate-depleted environments [11,12].

$$CH_4 + SO_4{}^{2-} \rightarrow HCO_3{}^- + HS^- + H_2O \ \Delta G^{0'} = -16.6 \ kJ \ mol^{-1} \ CH_4 \quad (1)$$

$$CH_4 + 8Fe(OH)_3 + 15H^+ \rightarrow HCO_3{}^- + 8Fe^{2+} + 21H_2O \ \Delta G^{0'} = -572 \ kJ \ mol^{-1} \ CH_4 \quad (2)$$

**Geochemical zoning**

**Pathway and stoichiometry of methane oxidation in sediments**

$$2O_2 + CH_4 \rightarrow CO_2 + 2H_2O$$

$$4NO_3{}^- + CH_4 \rightarrow 4NO_2{}^- + CO_2 + 2H_2O$$

$$8NO_2{}^- + 3CH_4 + 8H^+ \rightarrow 4N_2 + 3CO_2 + 10H_2O$$

$$SO_4{}^{2-} + CH_4 \rightarrow HS^- + HCO_3{}^- + H_2O$$

$$4MnO_2 + CH_4 + 7H^+ \rightarrow 4Mn^{2+} + HCO_3{}^- + 5H_2O$$

$$8FeOOH + CH_4 + 15H^+ \rightarrow 8Fe^{2+} + HCO_3{}^- + 13H_2O$$

$$8Fe(OH)_3 + CH_4 + 15H^+ \rightarrow 8Fe^{2+} + HCO_3{}^- + 21H_2O$$

$$4Fe_2O_3 + CH_4 + 15H^+ \rightarrow 8Fe^{2+} + HCO_3{}^- + 9H_2O$$

$$4Fe_3O_4 + CH_4 + 23H^+ \rightarrow 12Fe^{2+} + HCO_3{}^- + 13H_2O$$

**Three major pathways of methanogenesis in sediments**

$$CO_2 + 4H_2 \rightarrow CH_4 + 2H_2O$$

$$CH_3COOH \rightarrow CH_4 + CO_2$$

$$4CH_3OH \rightarrow 3CH_4 + CO_2 + 2H_2O$$

**Figure 1.** Pathways of $CH_4$ oxidation and three major pathways of methanogenesis (hydrogenotrophic, acetoclastic, and methylotrophic) in sediments. Among the materials involved, $O_2$, $NO_x{}^-$ ($NO_3{}^-$, $NO_2{}^-$), $SO_4{}^{2-}$, and $CO_2$ are dissolved in pore water, while $MnO_2$ and Fe (hydro)oxides in a solid state mixed with sediments. SMI: sulfate-methane interface.

Although sulfate-dependent AOM is considered to be the most significant process for methane consumption, especially for marine ecosystems, AOM via iron reduction is reported to occur in some marine and freshwater environments with depleted sulfate and nitrate [13]. Fe-AOM is of great significance in global methane sink, especially in marine sediments. However, its mechanism is still not well understood, possibly related to diagenesis, global $CH_4$ dynamics, and element cycling [14–17]. In this review, the studies pertaining to the conditions, microbiology, and geochemistry of Fe-AOM are summarized, and the environmental significance of Fe-AOM and challenges for future research are discussed.

## 1.1. Conditions for Fe-AOM in Marine Sediments

Although AOM coupled to sulfate reduction is considered to be common in marine sediments, a high rate of anaerobic methanotrophy is also observed in some marine anoxic sediments with no sulfate. It is also observed that the AOM rate is significantly higher than the rate of sulfate reduction in some environments, where AOM and sulfate reduction co-occur. These observations indicated the presence of other potential electron acceptors such as iron and manganese oxides [18,19]. However, as compared with sulfate-AOM,

the Fe-AOM process is less investigated because it is not as intensive as the sulfate-AOM. Nevertheless, the widespread presence of Fe-AOM can be indicated by biogeochemical profiling in marine sediments. The comprehensive inorganic data of the sediments collected from the Argentine Basin suggested that iron-driven AOM was most likely the major mechanism for iron reduction [20]. In another study of the Alaskan Beaufort Sea, correlation network analyses also suggested a conjugation of AOM to manganese or iron reduction in the sediment cores studied [21]. In the methanic zone of North Sea Helgoland mud, a strong correlation of anaerobic methane-oxidizing archaea (ANME) populations with $Fe^{2+}$ profile of pore water indicated the presence of iron-coupled AOM [22]. In deep Baltic Sea sediments below a shallow SMTZ, the elevated concentrations of dissolved $Fe^{2+}$ in pore water could be explained by methane oxidation with Fe oxides [23]. Recently combined geochemical and molecular evidence has revealed microbial iron reduction occurring in the deep methanic zone of the Mediterranean Sea, which might also be linked to a cryptic sulfur cycle and iron-coupled AOM [24].

In marine sediments, the natural conditions under which iron-coupled AOM happens are still poorly understood. The coexistence of pore water methane and sufficient reducible iron oxides, which might be the result of the high input of Fe oxides or transient diagenesis induced by organic matter and upward migration of methane, seems essential for the Fe-AOM process [12,20,25]. Rapid sediment accumulation prevents some iron oxides from being converted to authigenic iron sulfide, by which the preservation of iron oxides is further facilitated in the methanogenic zone below the sulfate–methane interface (SMI) [26]. In addition, as the result of transient diagenesis caused by disturbances such as climatic change or post-glacial sea-level rise, the iron oxide-rich deposits could be buried beneath sulfidic sediment layers [20,26,27].

Accordingly, a high concentration of $Fe^{2+}$ is frequently observed in the methanogenic zone below the SMTZ and is suggested to be produced by iron oxides coupled with AOM [24,28]. Crowe et al. found that the absence of sulfides allows $Fe^{2+}$ to accumulate at high concentrations in deep water [29]. Based on experimental results, Beal et al. reported that birnessite or ferrihydrite was involved in the AOM process in the absence of sulfate [11]. Modeling studies also supported the recognition that geochemical characteristics below the SMTZ, such as depleted sulfate, high concentrations of methane, and the availability of iron oxides, are the precedent condition for Fe-AOM [20]. It should be noticed that several studies discovered that iron oxides could be coupled to AOM, even in the sulfate-containing zone, although its mechanism is still enigmatic [11,20,30]. A study on an active methane seepage off Oregon showed the coexistence of methanogenesis, Fe-AOM, and sulfate reduction in the sediments. It was believed that the precipitation of iron sulfides, formed from reduced iron and sulfide in AOM processes, might accelerate sulfate-driven AOM [31].

### 1.2. Potential Forms of Iron Oxides for Fe-AOM

Because in situ observation of the Fe-AOM process in marine sediments is still challenging, not much is known about what form of iron oxide is acting as an electron acceptor for Fe-dependent AOM. In sediments, among iron oxides, potential electron acceptors include lepidocrocite, hematite, magnetite, ferrihydrite, goethite, akageneite, etc. [32]. Several methods can be applied to specify the forms of iron oxides in sediments, including sequential extraction, powdered X-ray diffraction, scanning electron microscope, etc. [33]. However, it is still difficult to pinpoint what form of iron oxide is involved in Fe-AOM in marine sediments. Regarding the transformation of iron oxides associated with Fe-AOM in sediments, most studies conducted are about the release of dissolved iron ($Fe^{2+}$) from sediment to pore water and the origin of reactive iron, i.e., $Fe(OH)_3$ [34]. It is found that the dissolved forms of Fe(III), such as Fe chelates or Fe complexes, are more active compared to solid minerals, but they are scarce under neutral conditions. Unfortunately, iron oxides generally appear in insoluble forms, so they cannot be easily used by microorganism cells. The bioavailability and reaction rate of Fe(III) reduction are often related to the form, solubility, stability, structure, and size of the iron oxides.

Ettwig et al. observed a much higher AOM activity stimulated by ferric citrate complex than nanoparticulate ferrihydrite via batch incubations [35]. Even among ferric complexes, each form is with a specific AOM-associated reductive activity. Scheller et al. found that as the terminal electron acceptor, ferric citrate-coupled AOM reached the highest activity, followed by iron EDTA, and ferric-NTA-coupled AOM had the lowest activity [36].

The crystallinity and conductivity of iron oxides also have a significant impact on Fe-AOM in iron oxide-rich sediments. Based on the experimental results, Nordi et al. reported that the iron-dependent AOM was more energetically favorable with ferric oxyhydroxide in amorphous than goethite as the electron acceptor [17]. It is speculated that it is more difficult for crystalline Fe-oxides to be biologically reduced by methane because of the kinetic limits, crystalline nature, or charges in surface structure due to adsorption of ions [37–40].

Additionally, the sizes of particle aggregates might influence the bioavailability of iron oxide minerals in microbial reduction, possibly because of their different spatial accessibility for microorganisms. It is also found that nanosized iron oxide aggregates, appearing in colloidal suspensions, could be reduced more rapidly by microorganisms by two orders of magnitude higher than macro-particulate forms [41].

### 1.3. Microbial Mechanisms

Marine microbial anaerobic methanotrophy is one of the key processes of mitigating methane emissions to the atmosphere. The sulfate-coupled AOM, by which the majority of methane is oxidized, is typically mediated by syntrophic microbial consortia of ANME and sulfate-reducing bacteria (SRB) [42]. ANME can be divided into three distinct phylogenetic clades called ANME-1 (with subgroups a and b), ANME-2 (with subgroups a, b, c and d), and ANME-3 [43]. ANME-1 is distantly related to the methanogenic orders Methanosarcinales and Methanomicrobiales, while both ANME-2 and ANME-3 are clustered within the Methanosarcinales [44]. Electrons are generated by ANMEs for SRB to reduce sulfates upon oxidizing methane by a reverse methanogenesis pathway.

In marine sediments, other available electron acceptors coupled to AOM, such as $NO_3^-$, $NO_2^-$, Mn(IV), and Fe(III), have also been identified. Large amounts of iron and other metal oxides are supplied into the sea by rivers from rock weathering. As major oxidized compounds in marine sediments, Fe oxides have great potential to serve as electron acceptors for AOM, especially when the sulfate is depleted or decoupled from the AOM process. It has been discovered that Fe-AOM is a prevalent biochemical process with complex mechanisms in marine environments, and different groups of microbes are involved. Several incubation studies with sediments collected from anoxic marine environments, especially from methane-rich environments, showed that the addition of Fe(III) oxides could enhance microbial AOM activity. Beal et al. found that microorganisms in marine methane seep sediments in the Eel River Basin are capable of oxidizing methane with manganese (birnessite) and iron (ferrihydrite) [11]. It is also observed that methane could be oxidized by deep-sea sediments added with artificial oxidants such as soluble ferric citrate and ferric-EDTA [36].

Since there is a lack of a pure culture of representative microbes, the microbial population responsible for Fe-AOM in marine sediments is still not well understood. However, several previous studies suspected some microbes possibly being related to Fe-AOM. After ten months of incubation of Eel River Basin sediments, the observed shift in the microbial community implied that ANME-1, ANME-3/*Methanococcoides* spp. might play a vital role in Fe-AOM with their metal-reducing bacteria partners (*Bacteroides*, *Desulfuromonas*, *Acidobacteria*, and *Verrucomicrobia*) [11]. A culturing study on the methane seep sediments of the Santa Monica Basin showed a high abundance of ANME-2a and ANME-2c, and a relatively low abundance of ANME-1 could be decoupled from their syntrophic SRB partners when ferric iron compounds were added [36]. Oni et al. observed that the distribution of the *Atribacteria*, methanogenic archaea, and *Methanohalobium*/ANME-3 related archaea are strongly correlated to the profile of dissolved $Fe^{2+}$ in sediments [22]. In addition, recent studies have suggested an ANME-2d-affiliated *Candidatus* Methanoperedens, which may

be a versatile methanotroph, is capable of using not only nitrate but also Fe oxidants as electron acceptors under different environmental conditions [35,45–48].

It was proposed that ANME groups associated with Fe-AOM could reduce ferric iron upon methane oxidation via the reverse methanogenesis pathway (Figure 2). In this process, electrons could be directly transferred to soluble metal ions or complexes or solid metal oxides. As with sulfate-AOM consortia, ANME could also reduce metal oxides by working with iron-reducing bacteria [36,49]. In natural settings to transfer an electron to solid ferric iron minerals, several strategies might be adopted by Fe-AOM associated microorganisms though these have not yet been proved. These strategies could be: (1) electron transfer by direct contact between cell and minerals; (2) indirect electron transfer by a metal chelate; (3) indirect electron transfer by redox-active organic molecules as electron shuttles such as multi-heme c-type cytochromes (MHCs) or humic acids; and (4) interspecies electron transfer by nanowires [36,50,51].

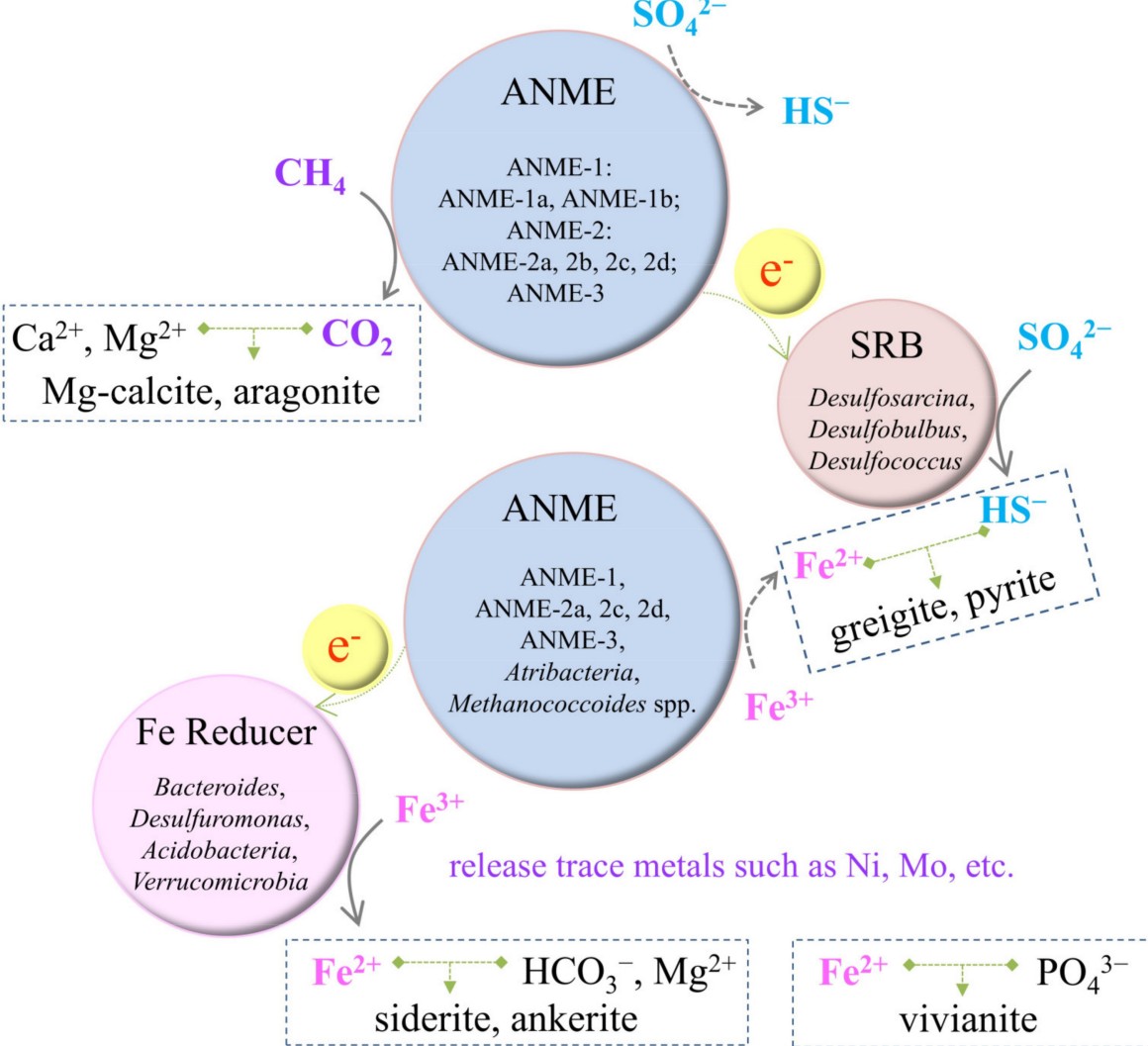

**Figure 2.** Possible interpretations for the biogeochemical process of Fe-AOM in sediments.

## 2. Geochemical and Environmental Significance

Fe-AOM is possibly a prevalent process in marine ecosystems which may play a significant role in regulating methane fluxes in deep-sea sediments. Moreover, the products of Fe-AOM, $Fe^{2+}$ and $CO_2$, could affect the geochemical cycles of sulfur and phosphorus as well as the formation of authigenic minerals. Thus, Fe-AOM might be a significant factor in global biogeochemical element cycling, especially in carbon balance.

### 2.1. Impact on Methane Dynamics

As the final product of the anoxic degradation of organic matter, methane is widespread in nature and commonly coexists with iron in deep marine sediments. In methane hydrate-bearing sediments, the pore water must be saturated with methane, creating a special geochemical environment. In the marine sediments rich with methane, the Fe-AOM process should be common, but its role in the global methane budget might have been underestimated previously [20]. Field investigation and model simulation revealed that Fe-AOM accounts for 9% of total $CH_4$ oxidation in comparison with 90% by sulfate and the rest by oxygen [26]. In addition, in the iron-rich marine environment of the early Archean period, 2.5 billion years ago, the possibility of a coupling of Fe(III) reduction with $CH_4$ oxidation theoretically should be an important carbon sink before the massive appearance of marine sulfate [52].

In the ecosystem with methane enrichment below the SMTZ, in which acetoclastic and hydrogenotrophic methanogenesis are deemed to be the main methanogenic processes [53], iron-reducing bacteria and sulfate-reducing bacteria could outcompete methanogens for substrates of acetate and hydrogen according to thermodynamic laws, resulting in the inhibition of acetoclastic, hydrogenotrophic methanogenesis, or both [1,54–56]. As a result, global methane cycling perhaps has been under the influence of the complex interaction among iron reduction, $CH_4$ production, or oxidation in deep marine sediments.

### 2.2. Impact on Iron Cycling

In consideration of the 8:1 Fe-$CH_4$ stoichiometry of the Fe-AOM reaction (Equation (2)), Fe-AOM likely has a greater impact on the cycling of iron over methane. Some diagenetic models and their applications in sediments are developed, suggesting that about 46% of $Fe(OH)_3$ reduction could be attributed to Fe-AOM [26].

The transformation between Fe(II)/Fe(III) plays a critical part in the geochemical iron cycle and the mineralization of organic substances (Figure 2). Various iron minerals such as pyrite, vivianite, siderite, and magnetite could be produced by the reaction of ferrous iron with carbonate, sulfide, phosphate, and residual Fe(III) [32,57,58]. Especially in seawater of the ancient Earth, rich in methane and ferrous iron, Fe-AOM may have potent influences on its composition [35,59]. Maerz et al. suggested that in Zambezi deep-sea fan sediments, Fe(II) phosphate vivianite is favorably produced [60]. Lim et al. observed a marked pyrite signature induced by AOM near the sulfate–methane interface [61].

### 2.3. Impact on Calcium, Sulfur, Phosphorus, and Other Elements

Iron-dependent pathways for methane oxidation generate a lot of research interests, not only because of their contribution to iron cycling and reducing methane emission from marine sediments to the atmosphere but also because of their potential influence on the geochemistry of marine carbonates, sulfur, phosphorus, or trace metal cycles via microbial metabolism or precipitation of mineral phases [12,25,26,60,62].

Sun et al. and Peng et al. reported that Fe-AOM plays a certain role in the formation of Fe-rich carbonate deposits, which is associated with cold seep activities and affects the distributions of goethite and carbonate in sediments [63,64]. In deeper sediment, the reactive Fe-containing mineral can be reduced via a sulfur cycle, related to the presence of $S^0$ and iron monosulfide minerals, or both [65,66]. In anoxic and sulfidic conditions, $Fe^{2+}$ could form iron sulfides (FeS and $FeS_2$), which are buried afterward in sediments permanently [67–69]. In the core sediments retrieved from the Western High

in the Sea of Marmara, a significant content of authigenic Fe(II) carbonate and greigite ($Fe_3S_4$) were observed [32]. Moreover, Fe-AOM is likely to have a considerable impact on the biogeochemical cycling of sedimentary phosphorus. Below the sulfidization front, high concentrations of dissolved ferrous iron lead to the sequestration of downward-diffusing phosphate as authigenic vivianite, resulting in a transient phosphorous accumulation directly below the sulfidization front [25,26,62]. Research on methane-rich sediments of the northern South China Sea suggested that vivianite formation below SMTZ could serve as a mineralogical marker of Fe-AOM, and almost half of the total reactive iron were sequestered by vivianite authigenesis according to the calculations [33].

Elevated $Fe^{2+}$ effluxes may have potentially significant impacts on the functioning of the marine ecosystem. For example, cyanobacteria blooms could be promoted by laterally transported iron due to Fe-AOM in surface sediments. Iron oxide-associated bio-essential elements, such as trace metals of Ni, Mo, Zn, etc., could be discharged and released to the environment during iron reduction [60,70,71]. Thereby, methanogenesis, a microbial process with imperative trace metal mobilization such as Ni, might be stimulated or enhanced by iron reduction in the deep methanic zone. Consequently, AOM-associated iron reduction is further promoted by this positive feedback (Figure 2).

## 3. Conclusions and Challenges

Methane is a potent greenhouse gas that is far more active than carbon dioxide and may enhance acidification in the marine environment [72]. Fe-AOM in the marine sediments has been identified by the elevated concentrations of dissolved iron in the methanic zone. Fe-AOM might play an important role in the capacity of $CH_4$ oxidation coupled to the reduction in ferric iron-bearing minerals by complex mechanisms mediated by relevant microorganisms. Fe-AOM could be one of the driving forces shaping the biosphere by affecting both the functioning of sediment ecosystems and the element cycling of marine carbonates, sulfur, phosphorus, or trace metals.

Since only limited investigations on Fe-AOM are conducted, the involving microbes and extracellular electron transfer mechanisms are largely undefined, rate-controlling factors are still unexplored, and many explanations of the observed phenomena are only informed speculations based on known mechanisms of other AOM processes and metal reduction. Novel strategies are therefore required to be implemented for better understanding the related process. It will be important to investigate the diversity and ecological distribution of responsible microorganisms in the marine environment, to pinpoint what iron speciation acting as the electron acceptor, to figure out the factors affecting marine Fe-AOM, to assess the contribution of Fe-AOM as a carbon sink on a global scale on both the ancient and the present Earth.

**Author Contributions:** Conceptualization, H.L.; writing—original draft preparation, H.Y. and S.Y.; writing—review and editing, H.Y., S.Y., H.L.; visualization, H.Y., S.Y., H.L.; supervision, H.L.; project administration, H.L.; funding acquisition, H.L. All authors have read and agreed to the published version of the manuscript.

**Funding:** This research was funded by China Geological Survey Project, grant number DD20190234.

**Institutional Review Board Statement:** Not applicable.

**Informed Consent Statement:** Not applicable.

**Data Availability Statement:** Not applicable.

**Conflicts of Interest:** The authors declare no conflict of interest.

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
