# Peer review of "Iron-Coupled Anaerobic Oxidation of Methane in Marine Sediments: A Review"

_jmse, doi:10.3390/jmse9080875_

Round 1
Reviewer 1 Report
In their manuscript, Hailin Yang and co-authors analyze information about Iron coupled anaerobic oxidation of methane in marine sediment. One major outcome of the study is that iron oxides is part of the major components in deep marine sediments and might play role as electron acceptors in AOM process.
By text. The name of phyla, genus must be written in italics
Page 1 L. 30 no reference.
Page 1 L. 38 In the presence of oxygen, methane is oxidized by widespread methanotrophic species of Alpha- and Gammaproteobacteria possessing the key enzyme methane monooxygenase. This is aerobic way oxidation of methane. In AOM oxygen (O2) cannot terminal electron acceptor.
Page 2 L. 50-56 Fe-AOM is characteristic not only for marine ecosystems, but also for freshwater lakes where iron and manganese were present, but depleted of sulphate and nitrate (Savvichev et al., 2017). Still, for marine ecosystems, the most significant sulfate-dependent AOM.
Page 2 L. 51-56 Missing reference on the articles
Figure 1 No decryption SMI
Figure 1 absent pathway of AOM with NO2- as electron acceptor
Figure 1 Why is there only one methanogenesis pathway in sediments? Three major pathways of methanogenesis are known: hydrogenotrophic, methylotrophic, and acetoclastic. And a new pathway involving of representatives Methanomassiliicoccales.
Page 2 L. 51 [5-11].
Page 4 L. 132 [30–33]
Page 5 L. 205 [1, 40–42]
Eq. 2 In manuscript Anna J. Wallenius et al., 2021 reaction of AOM with electron acceptor Fe(OH)3 have different values of energy, which not in manuscript Egger et al., 2015.
Page 4 L. 145 NO2- also
Page 5 L. 169 Oni et al.
Page 5 L. 170 Candidate division JS1 is Atribacteria now.
In section Microbial mechanisms absents information about ANME-2d (Candidatus Methanoperedens) as the main players not only in nitrate-dependent AOM, but iron-dependent anaerobic methane oxidation. This is considered in manuscripts Weber et al., 2017; Ettwig et al., 2016; Cai, C. et al., 2018; Roland et al., 2021 etc.)
Reference: Page 7 L. 283 after (Denmark) necessary deleted 1
Reference can be adjusted according to the instructions for authors.
Author Response
Response to the reviewer’s comments
Reviewer 1
By text. The name of phyla, genus must be written in italics.
[Answer] Thanks for pointing out this. All names of taxonomic units have been revised.
Page 1 L. 30 no reference.
[Answer] It has been added as to the suggestion, please see L.36.
Page 1 L. 38 In the presence of oxygen, methane is oxidized by widespread methanotrophic species of Alpha- and Gammaproteobacteria possessing the key enzyme methane monooxygenase. This is aerobic way oxidation of methane. In AOM oxygen (O2) cannot terminal electron acceptor.
[Answer] Electron acceptor O2 has been deleted, please see L.44.
Page 2 L. 50-56 Fe-AOM is characteristic not only for marine ecosystems, but also for freshwater lakes where iron and manganese were present, but depleted of sulphate and nitrate (Savvichev et al., 2017). Still, for marine ecosystems, the most significant sulfate-dependent AOM.
[Answer] Revised, please see L. 56-59. We do appreciate your suggestions which help us a lot in revising the manuscript.
Page 2 L. 51-56 Missing reference on the articles
[Answer] Thank you. The missing reference is added, please see L.59-61.
Figure 1 No decryption SMI
[Answer] SMI has been explained, please see L.69.
Figure 1 absent pathway of AOM with NO2- as electron acceptor
Figure 1 Why is there only one methanogenesis pathway in sediments? Three major pathways of methanogenesis are known: hydrogenotrophic, methylotrophic, and acetoclastic. And a new pathway involving of representatives Methanomassiliicoccales.
[Answer] Thank you very much. Figure 1 has been revised as suggested. Pathways of AOM with NOx- (both NO3- and NO2-) as the electron acceptor have been revised and added. Besides, three major pathways of methanogenesis are also mentioned.
Page 2 L. 51 [5-11].
Page 4 L. 132 [30–33]
Page 5 L. 205 [1, 40–42]
[Answer] Revised as suggested. Thank you.
Eq. 2 In manuscript Anna J. Wallenius et al., 2021 reaction of AOM with electron acceptor Fe(OH)3 have different values of energy, which not in manuscript Egger et al., 2015.
[Answer] Revised, please see L.55.
Page 4 L. 145 NO2- also
[Answer] NO2- is added, please see L.158.
Page 5 L. 169 Oni et al.
[Answer] Revised, please see L.182.
Page 5 L. 170 Candidate division JS1 is Atribacteria now.
[Answer] Revised, please see L.183.
In section Microbial mechanisms
absents information about ANME-2d (Candidatus Methanoperedens) as the main players not only in nitrate-dependent AOM, but iron-dependent anaerobic methane oxidation. This is considered in manuscripts Weber et al., 2017; Ettwig et al., 2016; Cai, C. et al., 2018; Roland et al., 2021 etc.)
[Answer] Candidatus Methanoperedens related information has been added, please see L.185.
Reference: Page 7 L. 283 after (Denmark) necessary deleted 1
[Answer] Thanks for pointing out this. 1 has been deleted.
Reference can be adjusted according to the instructions for authors.
[Answer] As to the suggestion, references have been changed.
Reviewer 2 Report
Although my review is ongoing, I hope my review report (though incomplete) may help the authors revise the manuscript.
Review comments on jmse-1316246-peer-review-v1
General comments
This manuscript is a mini-review of geochemical and microbiological studies on iron-coupled anaerobic oxidation of methane (Fe-AOM) in marine sediment.
Specific comments
L1 “Iron coupled …”
The authors’ used the hyphenated term “iron-coupled” in L45, which is the only case throughout the manuscript; other usages (6 times) are non-hyphenated “iron coupled”. Please choose either of them, hyphenated or non-hyphenated. I, personally, am inclined to choose the hyphenated one.
L9 “… (AOM) is one of the major processes for carbon sink”
The authors’ usage of the term “carbon sink” differs from other common usage. For example, Wikipedia describes “carbon sink” as “any reservoir, natural or otherwise, that accumulates and stores some carbon-containing chemical compound for an indefinite period and thereby lowers the concentration of CO2 from the atmosphere.” AOM is a process to generate CO2 via HCO3- and thus may be regarded as a carbon source. If the authors meant to take carbonate rocks (being formed via AOM) as a “carbon sink”, please describe so clearly.
L13 “… are more energetically favorable”
Please specify “more … favorable” than what.
L14 “a significant role as electron acceptor”
Please add “an” (indefinite article) before “electron acceptor”; please consult an English-editing company to check typos and grammatical errors throughout the manuscript.
L24 “Methane, just following carbon dioxide, is the second powerful greenhouse gas…”
Please clarify on which base “methane is second powerful.” Based on contribution to global warming, water vapor is the first powerful, CO2 is the second, and methane is the third. Based on global warming potential (*), methane is much more powerful than CO2.
* https://en.wikipedia.org/wiki/Global_warming_potential
L25 “(Methane) … playing an important role in marine and atmospheric chemistry”
Please be more descriptive to give enough information to state “playing an important role”; just saying “an important role” is not enough.
L28-30 “up to 90% of the methane is consumed by the anaerobic oxidation of methane (AOM) process conducted by microorganisms …”
This is an important part of the introduction of this manuscript and thus needs citation of relevant publications. In addition, AOM occurring in water column (oxygen minimum zones, OMZ) is also “a substantial methane sink” (**) and thus should be argued here in quantitative comparison with AOM in sediment.
** https://aslopubs.onlinelibrary.wiley.com/doi/abs/10.1002/lno.11235
L38 “(terminal) electron acceptors coupled to AOM, such as O2, NO3−, NO2− and …”
Please shape the sentence that deals with AOM (anaerobic = anoxygenic) and O2 simultaneously.
Author Response
Response to the reviewer’s comments
Reviewer 2
Specific comments
L1 “Iron coupled …”
The authors’ used the hyphenated term “iron-coupled” in L45, which is the only case throughout the manuscript; other usages (6 times) are non-hyphenated “iron coupled”. Please choose either of them, hyphenated or non-hyphenated. I, personally, am inclined to choose the hyphenated one.
[Answer] Thanks for pointing out this. The “iron-coupled” has been used.
L9 “… (AOM) is one of the major processes for carbon sink”
The authors’ usage of the term “carbon sink” differs from other common usage. For example, Wikipedia describes “carbon sink” as “any reservoir, natural or otherwise, that accumulates and stores some carbon-containing chemical compound for an indefinite period and thereby lowers the concentration of CO2 from the atmosphere.” AOM is a process to generate CO2 via HCO3- and thus may be regarded as a carbon source. If the authors meant to take carbonate rocks (being formed via AOM) as a “carbon sink”, please describe so clearly.
[Answer] Thank you. In the review article “Oceanic Methane Biogeochemistry”, Reeburgh 2007 has compiled the estimates of ocean methane sources and sinks, methane standing stock, and turnover times derived from a handful of rate measurements, the data calculation indicated that AOM is the major carbon sink of marine ecosystems. Besides, new techniques involving gene probes, determination of isotopically depleted biomarkers, and 14C-CH4 measurements showing that methane geochemistry in anoxic basins is dominated by seeps providing fossil methane. The role of anaerobic oxidation of methane has changed from a controversial curiosity to a major sink in anoxic basins and sediments. And this perspective on methane sources and the extent of methane oxidation has been widely accepted by the researchers’ related fields.
For example, a recent research carried by Yang et al., 2020 also mentioned that “the AOM process is also a major sink of the oceanic methane budget, consuming up to 300 Tg methane per year, equivalent to ~88% of the methane released from subsurface reservoirs”. Please see the detailed information from the links below.
https://pubs.acs.org/action/doSearch?field1=Contrib&text1=William+S.++Reeburgh
https://www.nature.com/articles/s41467-020-17860-8
L13 “… are more energetically favorable”
Please specify “more … favorable” than what.
[Answer] Thank you. The sentence has been modified to “However, other terminal electron acceptors of AOM, such as NO3−, NO2−, Mn(IV), Fe(III), are more energetically favorable than SO42−.” Please see L.12-13.
L14 “a significant role as electron acceptor”
Please add “an” (indefinite article) before “electron acceptor”; please consult an English-editing company to check typos and grammatical errors throughout the manuscript.
[Answer] Revised as suggested. Thank you very much.
L24 “Methane, just following carbon dioxide, is the second powerful greenhouse gas…”
Please clarify on which base “methane is second powerful.” Based on contribution to global warming, water vapor is the first powerful, CO2 is the second, and methane is the third. Based on global warming potential (*), methane is much more powerful than CO2.
* https://en.wikipedia.org/wiki/Global_warming_potential
[Answer] Thank you. Because the spatial and temporal distributions of water vapor and ozone vary greatly, these two gases are generally not taken into consideration in the decrement measures. According to the international treaty which extended the 1992 United Nations Framework Convention on Climate Change (UNFCCC), the “Kyoto Protocol” had defined six major long-lived greenhouse gases (GHG) includes carbon dioxide (CO2), methane (CH4), nitrous oxide (N2O), chlorofluorocarbons (CFCs), and carbon tetrachloride (CCI4). Please see the link below:
The sentence has been modified to “Methane, just following carbon dioxide, is the second powerful anthropogenic greenhouse gas in the atmosphere.” Please see L.23-24.
L25 “(Methane) … playing an important role in marine and atmospheric chemistry”
Please be more descriptive to give enough information to state “playing an important role”; just saying “an important role” is not enough.
[Answer] Thank you for pointing out this. More detailed information has added, please see L.25-31.
L28-30 “up to 90% of the methane is consumed by the anaerobic oxidation of methane (AOM) process conducted by microorganisms …”
This is an important part of the introduction of this manuscript and thus needs citation of relevant publications. In addition, AOM occurring in water column (oxygen minimum zones, OMZ) is also “a substantial methane sink” (**) and thus should be argued here in quantitative comparison with AOM in sediment.
** https://aslopubs.onlinelibrary.wiley.com/doi/abs/10.1002/lno.11235
[Answer] The references have been added as to the suggestion. We also emphasize that field investigation and model simulation revealed Fe-AOM accounts for 9% of total CH4 oxidation in comparison with 90% by sulfate and the rest by oxygen.
L38 “(terminal) electron acceptors coupled to AOM, such as O2, NO3−, NO2− and …”
Please shape the sentence that deals with AOM (anaerobic = anoxygenic) and O2 simultaneously.
[Answer] Thank you. Electron acceptor O2 has been deleted. We do appreciate your suggestions which help us a lot in revising the manuscript.
Reviewer 3 Report
Dear Authors,
Your manuscript, jmse-1316246 is a consist summary of current knowledge about iron coupled anaerobic methane oxidation in marine environments. It is worth publishing in JMSE but after consideration some suggestions listed below.
Line 75
Please explain abbreviation before it first use.
Line 90
Change the order of cited position.
Line 164
Please check the record the name of taxonomic units in whole manuscript, also figure 2. Use italic but in proper ways, for taxonomic names.
Line 170
The name of domain write with capital letter. The name of taxonomic unites write with italics.
Section: 1.3. Microbial mechanisms
Some short explanation/characteristic/differences for anaerobic methane oxidations clades is needed (e.g. ANME-1, ANME-2 etc.)
Author Response
Response to the reviewer’s comments
Reviewer 3
Line 75 Please explain abbreviation before it first use.
[Answer] Thank you very much. Revised as suggested. Please see L.84.
Line 90 Change the order of cited position.
[Answer] Thank you. It has been revised. Please see L.95.
Line 164
Please check the record the name of taxonomic units in whole manuscript, also figure 2. Use italic but in proper ways, for taxonomic names.
[Answer] All names of taxonomic units have been revised. Changes have been made throughout the manuscript as to the suggestion.
Line 170
The name of domain write with capital letter. The name of taxonomic unites write with italics.
[Answer] It has been changed because “Candidate division JS1” is “Atribacteria” now. Please see L.183. Thank you.
Section: 1.3. Microbial mechanisms
Some short explanation/characteristic/differences for anaerobic methane oxidations clades is needed (e.g. ANME-1, ANME-2 etc.)
[Answer] The detailed information of ANME clades are added, please see L.152-156. We appreciate your suggestions which help us a lot in improving our manuscript.
Round 2
Reviewer 2 Report
Review comments on jmse-1316246-peer-review-v2
General comments
The manuscript has been well revised and close to qualification for publication in JMSE, though
there are a few points to be edited or improved.
Specific comments
L9 “… (AOM) is one of the major processes for carbon sink”
[My first-round comment]
The authors’ usage of the term “carbon sink” differs from other common usage. For example,
Wikipedia describes “carbon sink” as “any reservoir, natural or otherwise, that accumulates and
stores some carbon-containing chemical compound for an indefinite period and thereby lowers the
concentration of CO2 from the atmosphere.” AOM is a process to generate CO2 via HCO3
- and thus
may be regarded as a carbon source. If the authors meant to take carbonate rocks (being formed via
AOM) as a “carbon sink”, please describe so clearly.
[Authors’ first-round response]
… In the review article “Oceanic Methane Biogeochemistry”, Reeburgh 2007 has compiled the
estimates of ocean methane sources and sinks … a recent research carried by Yang et al., 2020 also
mentioned that “the AOM process is also a major sink of the oceanic methane budget…
[My second-round comment]
Both Reeburgh 2007 and Yang et al., 2020 mentioned “methane sink”, not “carbon sink”. Please
clarify the distinction between “methane sink” and “carbon sink” to choose the more appropriate
terminology
L25-26 “GWP (global 25 warming potential)”
Abbreviations are usually placed in parentheses, aren’t they?
L232 and Figure 2 “siderite” “ankerite”
If Fe-AOM serves as an important process of “carbon source”, the process should include the
formation of authigenic carbonate rocks such as siderite FeCO3 and ankerite Ca(Fe,Mg,Mn)(CO3)2.
If the authors discuss “carbon sink”, please focus more on these iron-carbonate rocks in the main
text.
In addition, are there any publications reporting the occurrences of siderite and/or ankerite in the AOM-occurring sediment?
Author Response
Response to the reviewer’s comments
Reviewer 2 R2
L9 “… (AOM) is one of the major processes for carbon sink”
[My first-round comment]
The authors’ usage of the term “carbon sink” differs from other common usage. For example, Wikipedia describes “carbon sink” as “any reservoir, natural or otherwise, that accumulates and stores some carbon-containing chemical compound for an indefinite period and thereby lowers the concentration of CO2 from the atmosphere.” AOM is a process to generate CO2 via HCO3 - and thus may be regarded as a carbon source. If the authors meant to take carbonate rocks (being formed via AOM) as a “carbon sink”, please describe so clearly.
[Authors’ first-round response]
… In the review article “Oceanic Methane Biogeochemistry”, Reeburgh 2007 has compiled the estimates of ocean methane sources and sinks … a recent research carried by Yang et al., 2020 also mentioned that “the AOM process is also a major sink of the oceanic methane budget…
[My second-round comment]
Both Reeburgh 2007 and Yang et al., 2020 mentioned “methane sink”, not “carbon sink”. Please clarify the distinction between “methane sink” and “carbon sink” to choose the more appropriate terminology
[Answer] After careful discussion, we agree with the reviewer’s suggestion that our previous usage of “carbon sink” is not accurate. The sentence has been changed accordingly to “Anaerobic oxidation of methane (AOM) is one of the major processes to oxidize the methane in marine sediments.”
L25-26 “GWP (global warming potential)”
Abbreviations are usually placed in parentheses, aren’t they?
[Answer] Thanks for pointing out this. It has been changed to “global warming potential”.
L232 and Figure 2 “siderite” “ankerite”
If Fe-AOM serves as an important process of “carbon source”, the process should include the formation of authigenic carbonate rocks such as siderite FeCO3 and ankerite Ca(Fe,Mg,Mn)(CO3)2.
If the authors discuss “carbon sink”, please focus more on these iron-carbonate rocks in the main text. In addition, are there any publications reporting the occurrences of siderite and/or ankerite in the AOM-occurring sediment?
[Answer] Yes, thank you very much. The produced CO32− and HCO3− from methane oxidation may migrate to shallower sediments and react with Ca2+ and/or Fe2+ to form carbonates as calcite, ankerite, and/or siderite.
There are some publications reporting the occurrences of siderite and/or ankerite in the AOM-occurring sediments. Please see the following links:
https://www.sciencedirect.com/science/article/pii/S0016703717306920
https://www.sciencedirect.com/science/article/pii/S0967064517301339
